# Pneumococcal Surface Protein A-Hybrid Nanoparticles Protect Mice from Lethal Challenge after Mucosal Immunization Targeting the Lungs

**DOI:** 10.3390/pharmaceutics14061238

**Published:** 2022-06-11

**Authors:** Douglas Borges de Figueiredo, Kan Kaneko, Tasson da Costa Rodrigues, Ronan MacLoughlin, Eliane Namie Miyaji, Imran Saleem, Viviane Maimoni Gonçalves

**Affiliations:** 1Laboratório de Desenvolvimento de Vacinas, Instituto Butantan, São Paulo 05503-900, Brazil; actaestfabulla@gmail.com; 2Programa de Pós-Graduação Interunidades em Biotecnologia, Universidade de São Paulo, São Paulo 05508-070, Brazil; tasson.c.rodrigues@gmail.com; 3School of Pharmacy and Biomolecular Sciences, Liverpool John Moores University, Liverpool L3 3AF, UK; kankaneko@gmail.com; 4Laboratório de Bacteriologia, Instituto Butantan, São Paulo 05503-900, Brazil; eliane.miyaji@butantan.gov.br; 5Research and Development, Science and Emerging Technologies, Aerogen, IDA Business Park, H91 HE94 Galway, Ireland; rmacloughlin@aerogen.com

**Keywords:** hybrid nanoparticles, antigen delivery system, *Streptococcus pneumoniae*, vaccine, chitosan, PLGA, mucosal immunization

## Abstract

Pneumococcal disease remains a global burden, with current conjugated vaccines offering protection against the common serotype strains. However, there are over 100 serotype strains, and serotype replacement is now being observed, which reduces the effectiveness of the current vaccines. Pneumococcal surface protein A (PspA) has been investigated as a candidate for new serotype-independent pneumococcal vaccines, but requires adjuvants and/or delivery systems to improve protection. Polymeric nanoparticles (NPs) are biocompatible and, besides the antigen, can incorporate mucoadhesive and adjuvant substances such as chitosans, which improve antigen presentation at mucosal surfaces. This work aimed to define the optimal NP formulation to deliver PspA into the lungs and protect mice against lethal challenge. We prepared poly(glycerol-adipate-co-ω-pentadecalactone) (PGA-co-PDL) and poly(lactic-co-glycolic acid) (PLGA) NPs using an emulsion/solvent evaporation method, incorporating chitosan hydrochloride (HCl-CS) or carboxymethyl chitosan (CM-CS) as hybrid NPs with encapsulated or adsorbed PspA. We investigated the physicochemical properties of NPs, together with the PspA integrity and biological activity. Furthermore, their ability to activate dendritic cells in vitro was evaluated, followed by mucosal immunization targeting mouse lungs. PGA-co-PDL/HCl-CS (291 nm) or CM-CS (281 nm) NPs produced smaller sizes compared to PLGA/HCl-CS (310 nm) or CM-CS (299 nm) NPs. Moreover, NPs formulated with HCl-CS possessed a positive charge (PGA-co-PDL +17 mV, PLGA + 13 mV) compared to those formulated with CM-CS (PGA-co-PDL −20 mV, PLGA −40 mV). PspA released from NPs formulated with HCl-CS preserved the integrity and biological activity, but CM-CS affected PspA binding to lactoferrin and antibody recognition. PspA adsorbed in PGA-co-PDL/HCl-CS NPs stimulated CD80+ and CD86+ cells, but this was lower compared to when PspA was encapsulated in PLGA/HCl-CS NPs, which also stimulated CD40+ and MHC II (I-A/I-E)+ cells. Despite no differences in IgG being observed between immunized animals, PGA-co-PDL/HCl-CS/adsorbed-PspA protected 83% of mice after lethal pneumococcal challenge, while 100% of mice immunized with PLGA/HCl-CS/encapsulated-PspA were protected. Therefore, this formulation is a promising vaccine strategy, which has beneficial properties for mucosal immunization and could potentially provide serotype-independent protection.

## 1. Introduction

*Streptococcus pneumoniae* is a bacteria pathogen and the leading cause of lower respiratory tract infections [1]. Despite the existence of vaccines, the global burden of pneumococcal diseases remains high, and new vaccines with greater efficacy are needed [2]. Pneumococcal pneumonia causes significant morbidity, accounting for 5–15% of pneumonia cases in the US [3] and 19% in Europe [4]. The estimated 30-day mortality of invasive pneumococcal diseases (IPD) is 16.7% and can reach 26.5% in patients with meningitis plus other IPD [5]. Pneumococcal diseases have a significant impact on health-related quality of life [6]. All current vaccines are based on the protection offered by a few capsular polysaccharides from prevalent serotypes. However, there are more than 100 known serotypes of *S. pneumoniae* [7], and serotype replacement has occurred after the introduction of pneumococcal conjugate vaccines (PCV), due to the selective pressure caused by vaccination. For instance, serotype 14 was the most common in children before vaccination in Brazil, while after PCV introduction, serotype 3 became the most frequent in children aged <5 years, followed by serotypes 19A, 6A, 12F, and 6C, and serotypes 12F, 3, 8 and 9N in ≥5 years [8]. These phenomena mitigate the benefits of the vaccines [9], and hence, recombinant pneumococcal protein antigens are considered promising candidates for new serotype-independent vaccines. However, several protein-based or whole-cell vaccines have failed to progress in clinical trials, highlighting the importance of exploring new approaches, especially those that have the potential to block pneumococcal carriage [10], such as mucosal immunization.

To provide an adequate immune response, protein antigens are often formulated into a delivery system crucial for mucosal immunization. The lung mucosa is a very promising immunization route for respiratory pathogens such as pneumococcus, since mucosal vaccines could offer local and systemic immune responses and protect against pneumococcal colonization at the body entry sites [11]. Among the delivery systems, nanoparticles (NPs) are especially attractive for mucosal immunization because they can protect the antigen from degradation, act as an adjuvant, and enhance the uptake by antigen presenting cells (APCs) [12,13]. Polymeric NPs have the advantages of biocompatibility and versatility to be formulated in different sizes and shapes, and incorporate other molecules besides the antigen, which confer additional characteristics to the particles; for example, mucoadhesive substances such as chitosans to improve residence time and antigen presentation at mucosal surfaces and to facilitate the crossing of tight gap junctions between epithelial cells [14,15]. Chitosans also have intrinsic adjuvant properties, which can activate the cGAS–STING pathway and NLRP3 inflammasome [16]. Hence, an effective delivery system should guarantee the antigen presentation to the immune system and stimulate innate and adaptive immune responses.

APCs have a central role in the initiation of the innate and adaptive immune responses, and dendritic cells (DCs) are considered the main population of APCs. DCs exhibit high phagocytic capacity and are involved in the initiation of the adaptive immune responses that lead to memory and protection. Polymeric NPs and chitosans have intrinsic properties that stimulate immunogenicity through the uptake of NPs by DCs, which can occur through all four endocytosis pathways [17]. The interactions between NPs and DCs depend on the NP physicochemical properties, such as the surface chemistry [18,19] and the particle size [20,21]. The stimulation of DCs by NPs ultimately leads to metabolic changes and upregulation of costimulatory molecules, such as CD80, CD86, CD40, and MHC class II molecules, as well as the production of cytokines, which serve to stimulate immature T cells during antigen presentation [18,21]. Hybrid polymeric NPs containing chitosan are promising delivery systems for mucosal immunization with pneumococcal proteins, and DC activation assays could be employed to select formulations for protection evaluation in animal models.

Pneumococcal surface protein A (PspA) is a major antigen that is relatively conserved among the serotypes and involved in the evasion of pneumococci from host defenses, by inhibiting complement deposition and phagocytosis and binding the human bactericidal protein apolactoferrin [22]. Despite some sequence variation among PspA from different clinical isolates [23], some recombinant PspA fragments, such as PspA4Pro, induce strong cross-reactive protection against lethal pneumococcal challenge in mice [24]. PspA has recently been employed for nasal immunization using delivery systems such as nanogel [25], bacterium-like particles [26], polysorbitol transporter [27], liposomes [28], and protein bodies [29], showing protection in different animal models. In addition, we have demonstrated that mice immunized with adsorbed PspA4Pro onto polymeric NPs, such as poly(glycerol-adipate-co-ω-pentadecalactone) (PGA-co-PDL) NPs incorporated within l-leucine microparticles targeting the lungs [30], provided partial protection associated with IgG production in serum and lungs, and a rapid control of the infection after challenge [31]. Poly(lactic-co-glycolic acid) (PLGA) has also been investigated, which is a biocompatible copolymer already used in FDA-approved therapeutics, to formulate PspA4Pro NPs [32]. We have shown that hybrid NPs made from PLGA and chitosan exhibited a high antigen adsorption efficiency and immunogenicity in JAWSII DCs, indicating their suitability as a vaccine delivery vehicle [32].

Here, we compare PGA-co-PDL and PLGA NP formulations prepared by single or double emulsion/solvent evaporation method, to evaluate the influence of preparation conditions on NP characteristics. We also evaluate the effect of the incorporation of chitosan hydrochloride and carboxymethyl chitosan to obtain hybrid NPs with encapsulated or surface-adsorbed PspA4Pro, and analyze the integrity and biological activity of PspA4Pro released, to select the formulations that preserve PspA4Pro antigenicity. Finally, we select the formulations able to activate DCs in vitro, to immunize mice and evaluate the protection against lethal pneumococcal challenge.

## 2. Materials and Methods

### 2.1. Materials

Dichloromethane (DCM) was purchased from BDH laboratory supplies, Dorset, UK. Acetonitrile (HPLC grade), phosphate buffered saline (PBS, pH 7.4) tablets, poly(vinyl alcohol) (PVA, MW 9–10 KDa, 80% hydrolyzed), trifluoroacetic acid (TFA, HPLC grade), RPMI-1640 medium with l-glutamine and NaHCO_3_, Tween 20, Tween 80, alkaline phosphatase yellow (pNPP) liquid substrate buffer, and human lactoferrin were obtained from Sigma-Aldrich. l-leucine (l-Leu) was purchased from BioUltra, Sigma. Dimethyl sulfoxide (DMSO) and acetone were purchased from Fisher Scientific. Granulocyte macrophage colony-stimulating factor (GM-CSF) and interleukin 4 (IL-4) were purchased from Peprotech. A Micro BCA™ protein assay kit was purchased from Thermo Scientific. Polystyrene flat-bottom microplates (Costar 96-Well Microplates) were purchased from Cole-Parmer. Anti-PspA4 monoclonal antibody (monoclonal 22.003) was supplied by QED Bioscience, San Diego, CA, USA, and anti-mouse IgG conjugated to alkaline phosphatase by Jackson ImmunoResearch Europe Ltd., Cambridgeshire, UK. Skimmed milk and antibodies MHC Class II (I-A/I-E) FITC, CD11c PE-Cy7, CD11b APC-Cy7, CD80 PerCP-Cy5.5, CD86 PE, CD40 APC, FVS BV510 were supplied by BD Bioscience, San Diego, CA, USA. Carboxymethyl chitosan (CM-CS) (degree of deacetylation 80–95%, MW 30–500 KDa) and chitosan hydrochloride (HCl-CS) (degree of deacetylation 80–95%, MW 30–200 KDa) were purchased from Heppe Medical Chitosan GmbH, Halle, Germany. Poly(lactic-co-glycolic acid) (acid terminated, MW 7000–17,000) was purchased from Sigma, Welwyn Garden City, UK. Poly(glycerol-adipate-co-ω-pentadecalactone) (PGA-co-PDL) (ratio 1:1:1, glycerol, vinyl adipate and pentadecalactone, MW 15,700) was synthesized and characterized as described by Tawfeek et al. [33]. PspA4Pro with low content of endotoxin was prepared as described by Figueiredo et al. [34].

### 2.2. Preparation of NPs and Characterization

The NPs were prepared by two methods: the oil-in-water (*o*/*w*) single emulsion solvent evaporation method described by Kunda et al. [35], and (*w*/*o*/*w*) double emulsion/solvent evaporation method described by Alfagih et al. [36].

To verify the influence of organic solvent volume, PVA volume, PVA concentration, sonicator amplitude, and time of sonication on NP size, different conditions were evaluated for NP preparation using the *o*/*w* single emulsion method (Table 1). Briefly, 200 mg of PGA-co-PDL or PLGA was dissolved in DCM and probe sonicated upon addition to PVA (1st aqueous solution) for 2 min, to obtain an emulsion. This emulsion was added drop wise to the 2nd aqueous solution of PVA under magnetic stirring at a speed of 500 RPM for 90 min, at room temperature, to facilitate the evaporation of DCM. Then, the NPs were collected by centrifugation (78,000× *g*, 40 min, 4 °C).

For NP preparation by the *w*/*o*/*w* double emulsion method, 200 mg of PGA-co-PDL or PLGA were prepared with the same conditions chosen for the *o*/*w* single emulsion method (condition 10 for PGA-co-PDL: 2 mL DCM, 5 mL 10% PVA, 65% amplitude, 120 s, and 0.75% PVA, and condition 8 for PLGA: 2 mL DCM, 3 mL 10% PVA, 65% amplitude, 120 s, and 0.5% PVA), but including a probe sonication step (65% amplitude for 30 s) during the drop wise addition to the external aqueous solution containing PVA. The emulsion was kept under magnetic stirring at a speed of 500 RPM for 90 min, at room temperature, to facilitate the evaporation of DCM. Finally, the NPs were collected by centrifugation (78,000× *g*, 40 min, 4 °C).

To increase the adjuvant potential of NPs, two forms of chitosan were incorporated into the formulations: carboxymethyl chitosan (CM-CS) and chitosan hydrochloride (HCl-CS). To this end, 0.2 to 16 mg/mL of each chitosan was added to the second aqueous solution during NP preparation by the *o*/*w* single emulsion or the *w*/*o*/*w* double emulsion method. This procedure allowed chitosan to form bonds with the outer surface of the NPs to coat their surface. All factors investigated for NP optimization are summarized in Table 2.

Particle size, zeta-potential, and poly dispersity index (PDI) of all NPs were determined by Zetasizer Nano ZS (Malven Instruments Ltd., Malvern, UK), using 100 µL of NP suspended in 4 mL deionized water. From this suspension, 1 mL was added to the cuvette and measurement was performed in triplicates at 20 °C.

### 2.3. Protein Adsorption or Encapsulation

After obtaining the optimal NP formulations with and without chitosan, the next step was to incorporate PspA4Pro adsorbed on the surface or encapsulated into NPs, to verify the influence of the antigen location on the immune response. The selected NP suspensions (*o*/*w* single emulsion) were centrifuged (78,000× *g*, 40 min, 4 °C), and the resultant pellets were resuspended in vials containing 4 mL of PspA4Pro at a ratio of 25:1 *w*/*w* (NP: PspA4Pro) for protein adsorption. The resulting suspensions were left rotating on a HulaMixerTM Sample Mixer (Life Technologies, Invitrogen, UK) for 1 h at 20 RPM and room temperature. Two different pHs were tested, to improve protein loading. At pH 4.0 PspA4Pro is positively charged, which could improve the adsorption to negatively charged NPs, while the protein is negatively charged at pH 7.0. Antigen loading pHs are summarized in Table 2.

For encapsulation, PspA4Pro was added to the *w*/*o*/*w* double emulsion during the 1st sonication step, by adding the DCM dissolved polymer to a 10% *v*/*v* PVA solution containing PspA4Pro at a ratio of 10:2 *w*/*w* (NP: PspA4Pro). The effect of pH 4.0 on PspA4Pro encapsulation was not analyzed, to avoid submitting the protein to two potentially stressful conditions at the same time: low pH and exposure to the organic solvent of the 1st sonication step during NP preparation.

For both NP types, the amount of protein loaded in the NPs was determined by measuring the amount of protein remaining in the supernatant after centrifugation (78,000× *g*, 40 min, 4 °C) and calculating the difference between the initial amount and the amount in the supernatant. Hence, the non-loaded PspA4Pro was removed from the NPs by centrifugation prior to spray drying. PspA4Pro was measured by HPLC, as described by Kunda et al. [30], using PspA4Pro as a standard. The load efficiency (%LE) was calculated by Equation (1), and the amount (drug loading, *DL*) of PspA4Pro incorporated into the NP formulations was calculated using Equation (2).
(1)$$\%\mathrm{LE}=\frac{\left(initial\;PspA4Pro-supernantant\;PspA4Pro\right)}{initial\;PspA4Pro}\times 100$$
(2)$$DL=\frac{\left(initial\;PspA4Pro\;\left(\mu g\right)-supernantant\;PspA4Pro(\mu g\right))}{\;amount\;of\;NP\;\left(\mathrm{mg}\right)}$$

### 2.4. PspA4Pro In Vitro Release Studies

NPs (20 mg) were dispersed in 5 mL of PBS, pH 7.0. The samples were incubated at 37 °C and left to rotate for up to 24 h at 20 RPM on a HulaMixerTM Sample Mixer (Life Technologies, Invitrogen, UK). At time intervals of 30 min, 1 h, 2 h, 4 h, and 24 h, the samples were centrifuged (accuSpin Micro 17, Fisher Scientific, Loughborough, UK) at 17,000× *g* for 30 min. Then, 1 mL of the supernatant was removed and replaced with 1 mL PBS and incubated again. The supernatant was analyzed using the HPLC method, as mentioned above. Each experiment was repeated in triplicate, and the result was the mean value of three independently prepared samples (n = 3). The percentage of cumulative protein release was calculated using Equation (3).
(3)$$cumulative\;PspA4Pro\;release\;\left(\%\right)=\frac{cumulative\;PspA4Pro\;release}{PspA4Pro\;loaded}\times 100$$

### 2.5. PspA4Pro Integrity and Biological Activity

The integrity of PspA4Pro released from the NPs was determined using three methods. SDS-PAGE was employed to analyze if the primary structure of the protein was preserved. PspA4Pro recognition by antibodies was performed, to indicate if the 3D conformation of the protein was maintained. A lactoferrin binding assay determined if the biological activity was preserved, which is an additional indication that tertiary structure was not affected.

#### 2.5.1. SDS-PAGE

The size of PspA4Pro released from the NPs was determined by SDS-PAGE. SDS-PAGE was performed on a CVS10D omniPAGE vertical gel electrophoresis system (Geneflow Limited, Lichfield, UK) with 12% gel. Then, 15 µL of the supernatant released from each formulation in the in vitro release assay was applied to each lane, and PspA4Pro was employed as a control. The gel was run for approximately 2 h at a voltage of 100 V and stained with colloidal Coomassie blue.

#### 2.5.2. PspA4Pro Antigenicity

The antigenicity of PspA4Pro released from NPs was evaluated with an antibody recognition test. Briefly, a polystyrene flat-bottom 96-well microplate was coated with 5 µg of protein released from NPs dissolved in 100 µL sodium carbonate buffer per well. The microplate was incubated overnight at 4 °C, washed 3 times with PBS-T (PBS with 0.5% Tween 20), blocked with 200 µL 10% *w*/*v* skimmed milk in PBS, and incubated for 2 h at 37 °C. Then, the microplate was washed 3 times with PBS-T and 100 µL of anti-PspA4Pro monoclonal antibody, diluted 1:1000 to 1:64,000 in PBS-milk (5% *w*/*v* skimmed milk in PBS), was added to each well. The plate was incubated overnight at 4 °C, washed with PBS-T, and 100 µL of anti-mouse IgG conjugated to alkaline phosphatase was added to each well. The color was developed with 200 μL of alkaline substrate buffer, after 15 min at room temperature. The colorimetric changes were measured at 405 nm using a microplate reader (Epoch, BioTek Instruments Ltd., Stockport, UK). PspA4Pro (5 µg) was used as a positive control.

#### 2.5.3. Lactoferrin Binding Assay

PspA4Pro biological activity was determined by a lactoferrin binding assay. A polystyrene flat-bottom microplate of 96-wells was coated with 10 µg of human lactoferrin dissolved in 100 µL sodium carbonate buffer per well. The microplate was incubated overnight at 4 °C, washed 3 times with PBS-T, blocked with 200 µL 10% *w*/*v* skimmed milk in PBS, and incubated for 2 h at 37 °C. Then, the microplate was washed 3 times with PBS-T, and 5 µg of protein released from NPs was dissolved in 100 µL PBS-milk and added to each well. PspA4Pro (5 µg) was employed as a positive control, with PBS-milk as a negative control. The plate was incubated overnight at 4 °C, washed with PBS-T, and 100 µL of anti-PspA4Pro monoclonal antibody diluted 1:5000 in PBS-milk was added to each well. The microplate was incubated for 2 h at 37 °C, and the color was developed as described above for antibody recognition.

### 2.6. Nanocomposite Microparticles

NPs were incorporated into nanocomposite microparticles (NCMPs) by spray-drying, as previously described by Kunda et al. [30]. Briefly, NPs were suspended in l-leucine solution at a NPs-to-carrier ratio of 1:1.5 *w*/*w* and spray-dried at a feed rate of 10% with an atomizing air flow of 400 L/h, aspirator capacity of 100%, and an inlet temperature of 100 °C (outlet temperature of 45–47 °C). A Büchi B-290 mini spray-dryer (Büchi Labortechnik, Flawil, Switzerland) with a nozzle atomizer diameter of 0.7 mm and a high-performance cyclone (Büchi Labortechnik) were used to separate the dry particles (NCMPs) from the air stream. The NCMPs were collected and stored in a desiccator until further use.

### 2.7. Activation of DCs

Femurs of seven-week-old female specific-pathogen-free (SPF) BALB/c mice were used to collect bone marrow cells and differentiate DCs. Mice were euthanized with xylazine/ketamine solution (60 mg/kg of xylazine and 300 mg/kg of ketamine) for removing the femur. Each femur was sanitized with 70% *v*/*v* ethanol and transferred to RPMI medium containing 1000 IU/mL penicillin, 1000 µg/mL streptomycin, and 2.5 µg/mL amphotericin B (Invitrogen). After cutting femur extremities, cells were harvested with 3 mL of RPMI medium with GM-CSF (20 ng/mL) and IL-4 (10 ng/mL), the volume was completed to 10 mL and the cell suspension distributed between 2 Petri Optilux plates (100 × 20 mm, Falcon), an additional 5 mL of the same medium was added to each plate, and the plates were incubated at 37 °C and 5% CO_2_ for 3 days. Then, 10 mL of RPMI medium with GM-CSF and IL-4 was added to each plate, and the plates were incubated again for 4 days, and 106 cells per well were inoculated in a 24-well plate.

NCMP formulations, with or without chitosan, were added to each well. Empty NCMP, i.e., without PspA4Pro, were used as control of each formulation tested; with LPS as a positive control, and DCs without stimulus as a negative control. In addition, DCs without stimulus and without treatment were employed to calibrate the flow cytometer (FACSCanto, BD Biosciences). DCs were incubated with NCMP for 24 h at 37 °C and 5% CO_2_. Then, they were transferred to tubes with 4 mL of PBS, centrifuged (453× *g* for 10 min), washed twice with PBS and incubated for 30 min at 4 °C with the antibody mixture: MHC Class II (I-A/I-E) FITC, CD11c PE-Cy7, CD11b APC-Cy7, CD80 PerCP-Cy5.5, CD86 PE, CD40 APC, FVS BV510, and analyzed by flow cytometry. The gates applied to determine cell types are presented in Appendix A.

### 2.8. Mice Immunization

Six-to-seven-week-old female specific-pathogen-free (SPF) BALB/c mice (6 animals per group), obtained from the Medical School of the University of São Paulo (São Paulo, Brazil), were anesthetized with intraperitoneal (i.p.) inoculation of a xylazine/ketamine solution (20 mg/kg of xylazine and 100 mg/kg of ketamine). Mucosal immunization targeting the lungs was conducted as described by Rodrigues et al. [31]. For delivery of NCMPs (empty or containing 2 μg or 6 μg PspA4Pro), formulations were resuspended in saline (50 μL/dose) immediately before instillation into one nostril using a micropipette. This immunization procedure was previously shown to reach mice lungs [31]. Animals injected with saline or purified PspA4Pro subcutaneously (s.c.) (2 μg or 6 μg in 100 μL) and with purified PspA4Pro instilled into the lungs (2 μg or 6 μg in 50 μL, instillation into one nostril under anesthesia) were used as controls. Mice were immunized twice with a 14-days interval and challenged 21 days after the last immunization.

### 2.9. Measurement of Antibodies by Enzyme-Linked Immunosorbent Assay (ELISA)

ELISA was performed as described by Moreno et al. [24] using 96-well microplates coated with 5 μg/mL PspA4Pro. For the detection of serum antibodies, goat anti-mouse IgG conjugated with horseradish peroxidase (Sigma-Aldrich, Gillingham, UK) was used as secondary antibody. For IgG isotyping, goat anti-mouse IgG1, goat-anti-mouse IgG2a, and rabbit anti-goat IgG conjugated with horseradish peroxidase (Southern Biotech) were used. The titer was defined as the reciprocal of the highest dilution with an absorbance at 492 nm ≥ 0.1.

### 2.10. Pneumococcal Lethal Challenge

Twenty-one days after the last immunization, mice were anesthetized and challenged with 3 × 10^4^ CFU of *S. pneumoniae* strain ATCC6303, as described previously by Rodrigues et al. [31]. For the analysis of overall survival, mice were evaluated for up to 14 days after challenge. Animals were monitored twice daily after challenge, and lethargic animals with reduced ability to move were euthanized immediately with i.p. inoculation of a lethal dose of xylazine/ketamine solution (60 mg/kg of xylazine and 300 mg/kg of ketamine).

### 2.11. Statistical Analysis

Statistical analysis was performed using Prism 5.02 (GraphPad). Differences between groups were analyzed using one-way analysis of variance (ANOVA) with Tukey’s multicomparison test. Analysis of survival was performed using a Fisher exact test.

## 3. Results

### 3.1. Conditions for NP Preparation

We investigated ten different conditions, varying six parameters for NP preparation, to obtain the smallest particle size using the *o*/*w* single emulsion solvent evaporation method (Table 1). In general, the conditions that led to a NP size <200 nm were obtained with higher DCM volume, higher PVA concentration in both aqueous phases, and higher sonicator amplitude (conditions 2, 4, 7, 8, and 10, Table 1 and Figure 1a). These conditions also resulted in low PDI (≤0.1, Figure 1b). Given the smallest particle size and PDI, condition 10 (2 mL DCM, 5 mL 10% PVA, 65% amplitude, 120 s, and 0.75% PVA) was chosen for PGA-co-PDL formulations and condition 8 (2 mL DCM, 3 mL 10% PVA, 65% amplitude, 120 s, and 0.5% PVA) for PLGA formulations. These conditions were also applied to prepare PGA-co-PDL (condition 10) and PLGA (condition 8) NPs by the *w*/*o*/*w* double-emulsion solvent evaporation method. All PLGA NPs presented lower zeta potential than the PGA-co-PDL NPs prepared under the same conditions, while the size and PDI were very similar. All conditions yielded negatively charged NPs (Figure 1c).

### 3.2. Influence of Chitosan on NP Characteristics

Our previous results of mucosal immunization with PspA4Pro adsorbed PGA-co-PDL NPs showed the need for improving the immune response [19]. Thus, we evaluated the addition of HCl-CS or CM-CS to the NPs (Table 2). In general, both types of chitosan led to an increase of NP size. The PDI remained low for all formulations, but there was a tendency for increase, with the CM-CS concentration seen at 4, 6, and 8 mg/mL (Figure 2b). The addition of HCl-CS changed the zeta-potential, and NPs became positively charged, while the addition of CM-CS diminished the zeta-potential (Figure 2c). We defined 1 mg/mL of HCl-CS or CM-CS for PGA-co-PDL NP formulations using the *o*/*w* single emulsion method and used these concentrations to prepare PGA-co-PDL/chitosan hybrid NPs by the *w*/*o*/*w* double emulsion method. For PLGA NPs, we defined 1 mg/mL of HCl-CS or 0.5 mg/mL of CM-CS by the *o*/*w* single emulsion and used these concentrations to prepare PLGA/chitosan hybrid NPs by the *w*/*o*/*w* double emulsion method. The chitosan concentrations were chosen to yield the smallest particle sizes between both polymers used and with similar PDI. The increase of NP size was higher when the *w*/*o*/*w* double emulsion method was employed for both polymers, while the charge and PDI were similar (Table 3).

### 3.3. Protein Adsorption or Encapsulation

After defining the optimal NP formulations and chitosan concentration, the selected conditions were employed to prepare hybrid NPs with surface-adsorbed and encapsulated PspA4Pro using the *o*/*w* single and *w*/*o*/*w* double emulsion methods, respectively (Table 2). We evaluated the effect of pH and chitosan type on PspA4Pro adsorption and encapsulation (Table 4). All *o*/*w* single emulsion formulations performed at pH 7.0 had a higher loading efficiency (>90%) for adsorption of PspA4Pro onto hybrid NPs containing HCl-CS than CM-CS. As the NPs without chitosan or with CM-CS were negatively charged, adsorption of PspA4Pro was also carried out at pH 4.0, because at this pH PspA4Pro is positively charged (pI = 4.8). However, an increase in the loading efficiency at pH 4.0 was observed only for PLGA NPs (Table 4).

### 3.4. PspA4Pro In Vitro Release

PspA4Pro in vitro release was evaluated, and the lowest protein release (≤50%) in 24 h was observed when protein adsorption was achieved with PGA-co-PDL NP presenting opposite charges; i.e., at pH 4.0 for NPs without chitosan and hybrid NPs with CM-CS, and pH 7.0 for hybrid NPs with HCl-CS, while a release of between 70–80% was observed for formulations prepared with PspA4Pro and NP presenting the same charge (Figure 3a). In the case of PLGA NPs, the lowest protein release was achieved with PLGA hybrid NPs with CM-CS at both pH and with PLGA NPs prepared without chitosan at pH 7.0 (Figure 3b). When PspA4Pro was encapsulated into PGA-co-PDL NPs, the release was 80% without chitosan, 40% for hybrid NPs with CM-CS, and less than 20% with HCl-CS (Figure 3c). PLGA NPs with encapsulated PspA4Pro showed a release under 10% for all formulations (Figure 3d). However, for all NPs, PspA4Pro was rapidly released, reaching a plateau after 2 h from the beginning of the assay (Figure 3).

### 3.5. PspA4Pro Integrity and Biological Activity

The size of the PspA4Pro released from the NPs was determined by SDS-PAGE (Figure 4). No degradation of PspA4Pro was observed, but bands of approximately 100 kDa were detected (Figure 4). These bands were more distinct in samples from PspA4Pro encapsulated into hybrid NPs containing CM-CS (Figure 4b,d), and this could have been due to dimer or aggregate formation. All samples showed a band of the same size as the PspA4Pro control, which indicates that PspA4Pro integrity was preserved in terms of polypeptide primary structure. The apparent differences in protein content were due to the differences in release from NPs.

The antigenicity of PspA4Pro released from NPs was evaluated using an antibody recognition test. The presence of CM-CS strongly affected the antibody recognition of PspA4Pro released from NPs prepared with single and double emulsions (Figure 5). Considering values of absorbance at 405 nm above 0.1 as the threshold for antibody recognition, PspA4Pro released from NPs containing CM-CS was not detected with antibody dilutions greater than 1:8000. Furthermore, PspA4Pro released from NPs in which it was encapsulated with HCl-CS or without chitosan (Figure 5b,d) was recognized by the monoclonal antibody diluted >1:32,000 (Figure 5). This indicated that the PspA4Pro released from these formulations presented the expected conformation, which allowed the antibodies to bind.

Lactoferrin binding of PspA4Pro released from the NPs was evaluated as a proxy for its biological activity. Human lactoferrin is the natural ligand of PspA, and their interaction efficiently inhibits the bactericidal activity of lactoferrin on *Streptococcus pneumoniae* [37]. Lactoferrin binding also provides an indication of tertiary structure preservation. The presence of CM-CS in PspA4Pro-adsorbed and PspA4Pro-encapsulated NPs virtually abolished the protein’s ability to bind human lactoferrin, and preparation of NPs at pH 4.0 was also detrimental to PspA4Pro activity (Figure 6). The PspA4Pro released from all NPs prepared without chitosan exhibited reduced lactoferrin binding activity. However, the protein released from PspA4Pro-adsorbed NPs with HCl-CS at pH 7.0 exhibited the same lactoferrin binding activity as the control, and the protein released from PspA4Pro-encapsulated NPs with HCl-CS displayed a higher lactoferrin binding activity than the PspA4Pro standard (Figure 6).

### 3.6. Activation of DCs

DCs were differentiated from mouse bone marrow cells and stimulated in vitro with the NCMPs/PspA4Pro-adsorbed NPs with or without HCl-CS and NCMPs/PspA4Pro-encapsulated NPs with or without HCl-CS. Empty NCMPs/NPs (without PspA4Pro) were used as control. Differences in the percentage of CD40+ and MHC Class II (I-A/I-E)+ cells after stimulation with the distinct formulations were not statistically significant for NCMPs/PspA4Pro-adsorbed NPs (Appendix A). NCMPs/PspA4Pro-adsorbed PGA-co-PDL NPs containing HCl-CS were able to stimulate CD80+ and CD86+ cells (Figure 7a,b), while NCMPs/PspA4Pro-adsorbed PLGA NPs without chitosan only stimulated CD80+ upregulation on cells (Figure 7a). NCMPs/PspA4Pro-encapsulated PGA-co-PDL NPs did not activate DCs (Figure 7c,d and Appendix A). However, NCMPs/PspA4Pro-encapsulated PLGA NPs stimulated CD40+, CD80+, CD86+, and MHC-II (I-A/I-E)+ upregulation (Figure 7c,d and Appendix A). When comparing the pairs of NPs with or without HCl-CS, we can observe the adjuvant effect of chitosan. Hence, only NCMPs/PspA4Pro-adsorbed PGA-co-PDL NPs containing HCl-CS were able to activate DCs (Figure 7a,b). However, in NPs with PspAP4Pro encapsulated, it was the presence of HCl-CS in NCMPs/PspA4Pro-encapsulated PLGA NPs that stimulated CD80+, CD86, CD40+, and MHC II (I-A/I-E)+ cells (Figure 7c,d and Appendix A). Therefore, we selected NCMPs/PGA-co-PDL/HCl-CS PspA4Pro-adsorbed and NCMPs/PLGA/HCl-CS/PspA4Pro-encapsulated to immunize the mice.

### 3.7. Induction of Anti-PspA4Pro Antibodies by Immunization with NP/NCMPs

Mice received two doses of the formulations, and anti-PspA4Pro IgG titers were measured in the serum obtained from blood samples collected 14 days after each dose. There was a significant increase of IgG titers in the serum of mice immunized with PspA4Pro sc and NCMP/NP PspA4Pro compared with saline after two doses containing 2 µg (Figure 8a) or 6 µg (Figure 8b) PspA4Pro. Statically significant differences between IgG titers in serum of mice immunized with PspA4Pro sc and NCMP/NP PspA4Pro were observed only when the mice received two doses containing 6 µg PspA4Pro (Figure 8b). A negligible antibody response was observed in the group that received purified PspA4Pro or empty NPs into the lungs. No differences in IgG1/IgG2a ratios were observed between animals immunized with PspA4Pro sc and NCMPs/NPs PspA4Pro (Figure 8c,d).

### 3.8. Pneumococcal Lethal Challenge with Strain ATCC6303 (Serotype 3, PspA5)

The efficacy of lung immunization with NCMPs/PspA4Pro hybrid NPs with HCl-CS was then tested against a lethal challenge with pneumococcal strain ATCC6303, delivered to the lungs 21 days after the second dose. Partial protection was observed for mice immunized with PspA4Pro sc or NCMPs/PLGA/HCl-CS/PspA4Pro encapsulated (33.3%, 2 out of 6, *p* = 0.23, Fisher Exact Test) and NCMPs/PGA-co-PDL/HCl-CS/PspA4Pro adsorbed (67%, 4 out of 6, *p* = 0.03, Fisher Exact Test) when mice were immunized with two doses of 2 µg PspA4Pro (Table 4). When mice were immunized with two doses of 6 µg PspA4Pro, the protection observed for mice immunized with PspA4Pro sc remained the same, but it increased for mice receiving NCMPs/PGA-co-PDL/HCl-CS PspA4Pro-adsorbed NPs, reaching 83.3% (5 out 6, *p* = 0.007, Fisher Exact Test), and 100% for mice receiving NCMPs/PLGA/HCl-CS/PspA4Pro-encapsulated NPs (Table 5).

## 4. Discussion

The use of recombinant proteins, with NPs for adjuvant properties, has been gaining interest, due to the precision of the resulting immune response and the reduced potential for adverse effects. These NP formulations also have the potential to induce protective immune responses through mucosal routes such as the lungs [38]. The properties of the NP and the delivery of the antigen have a great influence on the successful induction of protective immune response, and this study investigated the potential of incorporating chitosan with PGA-co-PDL or PLGA polymeric NPs (hybrid NPs) with encapsulated or surface-adsorbed PspA4Pro for offering protection against pneumococcal disease.

PGA-co-PDL is a biodegradable polyester polymer synthesized by a lipase from *Candida albicans* and has free hydroxyl groups that can be eventually used for chemical modifications [21]. PLGA is a synthetic polylactide polymer hydrolyzed at very slow rate into lactic acid and glycolic acid, which are biologically compatible and easily metabolized [39]. To characterize and optimize NP formulations, we prepared PGA-co-PDL and PLGA NPs using the *o*/*w* single emulsion solvent evaporation method, using 10 different conditions for each polymer based on previous results of our group [35], and applied the conditions that produced the smallest particles (Figure 1a) to elaborate the *w*/*o*/*w* double-emulsion according to the previously published method [36]. All NPs were negatively charged and PDI remained low, demonstrating that the particles were quite homogenous in all formulations tested (Figure 1b). Despite being prepared with different polymers, particle size and PDI of PGA-co-PDL and PLGA NPs were similar for each condition evaluated, while PLGA NPs presented lower zeta potential values (Figure 1c), which correlate with PLGA chemical structure [39].

After defining the NP formulation conditions, two types of chitosan were incorporated into the NPs: HCl-CS or CM-CS. These chitosans exhibit high solubility and ease of handling compared to conventional chitosans, which require the use of low pH solvents for solubilization [32]. The HCl-CS changed the zeta potential of NPs, from negative to positive, while the CM-CS increased the negative charge. Both had little effect on PDI, and the particle size increased accordingly with concentration (Figure 2). The NP zeta potential is a very important characteristic, and it is generally accepted that positively charged NPs exhibit greater immunogenicity than those negatively charged, as there is greater interaction of the positive charged NPs with negatively charged mucosal tissues [15]. PGA-co-PDL and PLGA are hydrophobic and suffer from low encapsulation of hydrophilic actives such as proteins, hence the addition of HCl-CS [40]. The beneficial effect of HCl-CS was the increased PspA4Pro loading into NPs at pH 7.0 (Table 4), possibly due to electrostatic interactions, because at pH 7.0 PspA4Pro is net negatively charged (pI = 4.8) and the NPs containing HCl-CS are positively charged [32]. An attempt to increase the PspA4Pro loading via adsorption at pH 4.0 failed for PGA-co-PDL NP formulations with negative zeta potential, but there was an increase from 21% to 91% of the %LE for PLGA without chitosan, and from 26% to 74% for PLGA with CM-CS.

The *w*/*o*/*w* double emulsion method is the most common method for the encapsulation of water soluble compounds such as antigens [36], thus the NPs produced by this method were expected to encapsulate PspA4Pro, which could explain lower percentage of PspA4Pro released from encapsulated formulations compared to *o*/*w* single emulsion formulations with surface-adsorbed PspA4Pro (Figure 3). PspA4Pro integrity, recognition by antibody, and lactoferrin binding ability were analyzed in samples from the PspA4Pro in vitro release assay. The results showed that CM-CS had notable consequences for the PspA4Pro structure, which was not recognized by antibodies and did not bind lactoferrin (Figure 5 and Figure 6). The electrophoretic pattern of PspA4Pro released from the formulations containing CM-CS was also altered, and a band of approximately 100 kDa appeared (Figure 4). Although the reasons remain unclear, a possible explanation could be a strong interaction between CM-CS and PspA4Pro or aggregated PspA4Pro released from the formulations, which could impair the lactoferrin binding and the antibody recognition. The structural state of the protein antigen can influence recognition by the immune system [41], which encompasses multiple mechanisms by various immune cells [42]. Antigen recognition by B cell receptors is dependent on the original conformation of the antigen via complementarity-determining regions and highlights the significance of the antigen protein fold stability, which can influence the conformational epitopes and, ultimately, the recognition and response by the immune system [43]. Therefore, formulations with CM-CS were not used for further testing in mice.

APCs in the lung serve fundamental roles in the initiation of the innate and adaptive immune responses, with the lung DCs serving as an integral APC for initiating the adaptive immune responses [44]. The NCMPs/PspA4Pro-adsorbed NPs with or without HCl-CS and NCMPs/PspA4Pro-encapsulated NPs with or without HCl-CS were employed in the assay of DC activation, which was used to select the formulation for mice immunization, to reduce the number of animals in the experiment. The CD80 and CD86 are well recognized as important providers of co-stimulatory signals for further T cells activation [45]. The differences observed between the DC responses to adsorbed and encapsulated NPs reflect the potentially significant influence of the antigen incorporation approaches on the immune response. Each approach has advantages and disadvantages, which have been previously postulated in the literature. For example, entrapment of the antigen is theorized to offer protection from the external environment, but may result in an incomplete release [26], which may compromise the immune response. On the other hand, adsorption of the antigen may simplify particle production, but may result in high burst release, which can lead to deficient immune responses [46]. As seen in Figure 4, the release profiles of PspA4Pro from the adsorbed and encapsulated NPs may indeed follow these trends. The PspA4Pro adsorbed PLGA NP exhibits a greater release compared to the PGA-co-PDL counterpart and correspondingly exhibits lower immunogenicity. The higher immunogenicity of the PspA4Pro encapsulated PLGA NP could also be correlated with the lower antigen release compared to the PGA-co-PDL counterpart.

The adjuvant effect of chitosan was also verified in the DC activation assay. Chitosan and its derivatives are known for their immunostimulatory effects, which trigger DC maturation with upregulation of CD40, CD80, and CD86, and MHC II molecules [17]. Thus, the comparison of NCMPs/PGA-co-PDL/PspA4Pro-adsorbed NPs pairs with or without HCl-CS showed that only formulations containing chitosan stimulated CD80+ and CD86+ cells (Figure 7). Similarly, only NCMPs/PLGA/PspA4Pro-encapsulated NPs with chitosan upregulated all four markers of DC maturation (Figure 7 and Appendix A). From these results, two hybrid NPs were selected for animal immunization: PGA-co-PDL/HCl-CS/PspA4Pro-adsorbed and PLGA/HCl-CS/PspA4Pro-encapsulated.

To validate the effects of NCMPs/hybrid NPs, the formulations were investigated in vivo. NCMPs were produced, to overcome cold chain requirements and maintain the stability and integrity of the PspA4Pro [30]. Administration into the lungs of mice is suboptimal, and the nasal route is an alternative route for targeting the lungs of mice, and we have demonstrated that instillation of 50 μL Evans Blue into one nostril of anesthetized mice reached the lungs [31]. The lung presents a unique portal for exposure to the immune system, which enables the establishment of innate and adaptive immune responses. These responses not only provide local protection, but also confer systemic immunity that requires complex crosstalk between the innate and adaptive immune cells [46]. The results of antibody induction and protection against lethal challenge after 2 doses of NCMPs/NPs containing 2 μg or 6 μg of PspA4Pro confirmed that the activation of DCs is a useful predictor of the immune responses. Compared to our previous results using NCMPs/NPs prepared with PGA-co-PDL containing 2 μg PspA4Pro without chitosan [31]; here, we obtained the same level of protection, i.e., 67% survival after challenge with or without chitosan (Table 5). However, when the PspA4Pro dose was augmented to 6 μg, the antibody level increased by approximately one log (Figure 8) and the protection to 83% (5 out 6 mice survived after challenge, Table 5), indicating the adjuvant potential of PGA-co-PDL/HCl-CS NPs to stimulate systemic and local immune responses and protection against lethal challenge. The formulation using PLGA and HCl-CS to encapsulate PspA4Pro stimulated the highest activation levels of CD80+ and CD86+ cells (Figure 7c,d) and was also able to stimulate CD40+ and MHC-II (I-A/I-E)+ cells (Appendix A), which correlated to 100% protection of mice from the lethal challenge (Table 4). This observation is consistent with the role of CD40 and the corresponding association with CD86 and MHC II upregulation, which fulfil significant roles in the adaptive immune processes of T cell activation and antigen presentation, respectively [42].

A future direction could be to evaluate how the formulations affect the persistence of immunity, as well as the mechanisms of T and B cell responses. In addition, IgA levels could be determined for the evaluation of mucosal immunity in a further characterization of the immune response. Moreover, evaluation of the memory and persistence with regards to formulation characteristics would aide optimization.

## 5. Conclusions

We obtained optimal conditions for hybrid NP preparation using two biodegradable polymers (PGA-co-PDL and PLGA) incorporating two chitosan types (HCl-CS and CM-CS). Both polymers yielded hybrid NPs with similar sizes and PDI, but incorporation of HCl-CS led to positively charged NPs, and CM-CS to negatively charged NPs. CM-CS reduced the integrity and biological activity of PspA4Pro released, whereas this was preserved with HCl-CS. Moreover, the presence of HCl-CS was necessary to improve the activation of DCs in vitro. PLGA is commercially available, and it has the advantage of being already approved for pharmaceutical applications. Interestingly, PLGA formulations with or without HCl-CS presented one of the lowest PspA4Pro release in vitro, which could indicate the significance of antigen retention within the particles for optimal immune responses and could also explain the highest protective response observed compared to PGA-co-PDL/HCl-CS hybrid NPs. The results indicate that the NCMP/PLGA/HCl-CS hybrid NP formulation is a promising vaccine formulation strategy, which has beneficial properties as a dry powder and could potentially target a wide range of *S. pneumoniae* serotypes.

## Figures and Tables

**Figure 1 pharmaceutics-14-01238-f001:**
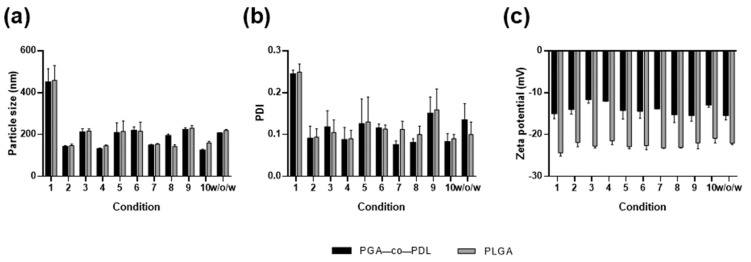
Analysis of particle size, poly dispersity index (PDI), and zeta potential of nanoparticles. (**a**) particle size, (**b**) PDI, and (**c**) zeta-potential. Conditions 1 to 10 are described in Table 1 to prepare NPs by *o*/*w* single emulsion solvent evaporation method, and *w*/*o*/*w* represents the values for NPs prepared by double-emulsion solvent evaporation method, using condition 10 for PGA-co-PDL and condition 8 for PLGA. Data represent mean ± SD, n = 3.

**Figure 2 pharmaceutics-14-01238-f002:**
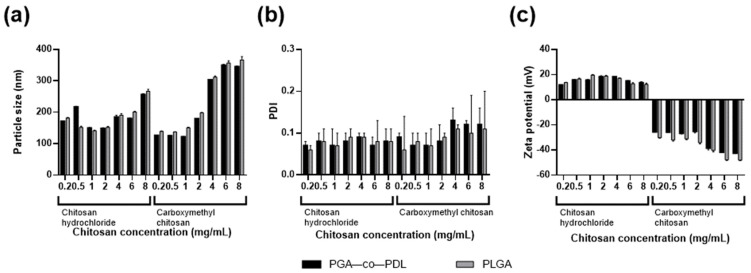
Effect of chitosan hydrochloride or carboxymethyl chitosan addition on particle size, poly dispersity index (PDI) and zeta-potential of hybrid NPs prepared by *o*/*w* single emulsion method. (**a**) particle size, (**b**) PDI, and (**c**) zeta-potential of PGA-co-PDL and PLGA hybrid nanoparticles. Data represent mean ± SD, n = 3.

**Figure 3 pharmaceutics-14-01238-f003:**
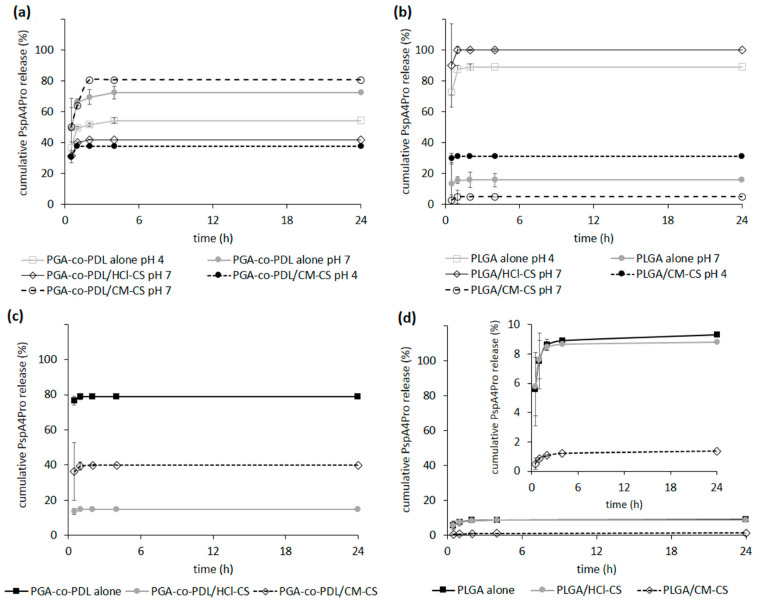
In vitro release of pneumococcal surface protein A (PspA4Pro) from NPs in phosphate buffered saline, pH 7.4. (**a**) PGA-co-PDL and (**b**) PLGA NPs carrying surface-adsorbed PspA4Pro at pH 4 or 7 prepared by *o*/*w* single emulsion. (**c**) PGA-co-PDL and (**d**) PLGA NPs prepared at pH 7.0 by *w*/*o*/*w* double emulsion with encapsulated PspA4Pro. The inset in (**d**) shows the same data in another Y-axis scale. Hybrid NPs were prepared with carboxymethyl chitosan (CM-CS) and with chitosan hydrochloride (HCl-CS). Data represent mean ± SD, n = 3.

**Figure 4 pharmaceutics-14-01238-f004:**
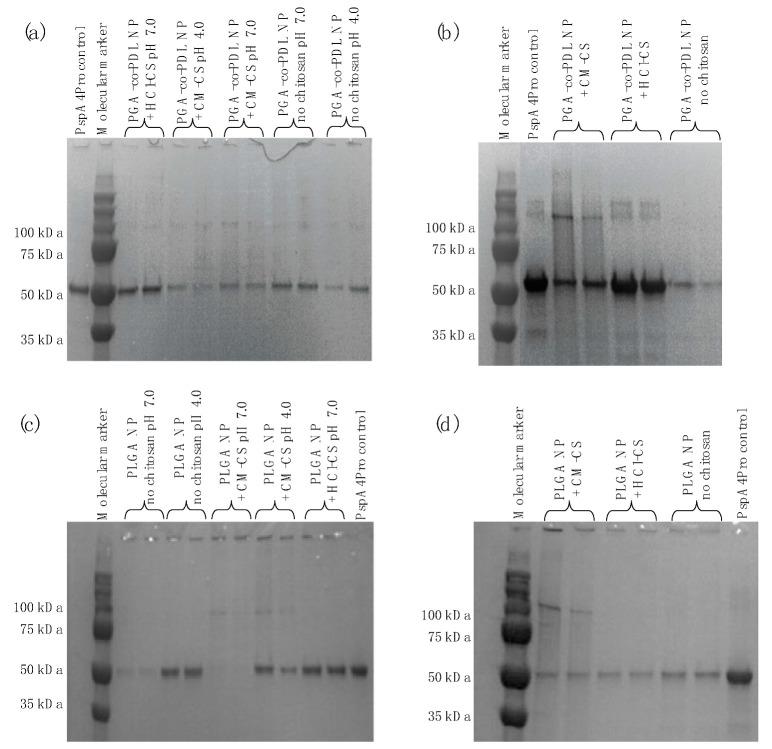
SDS-PAGE of PspA4Pro released from PGA-co-PDL or PLGA NPs. (**a**) PGA-co-PDL NPs carrying surface-adsorbed PspA4Pro prepared by *o*/*w* single emulsion. (**b**) PGA-co-PDL NPs prepared at pH 7.0 by *w*/*o*/*w* double emulsion with encapsulated PspA4Pro. (**c**) PLGA NPs carrying surface-adsorbed PspA4Pro prepared by *o*/*w* single emulsion. (**d**) PLGA NPs prepared at pH 7.0 by *w*/*o*/*w* double emulsion with encapsulated PspA4Pro.

**Figure 5 pharmaceutics-14-01238-f005:**
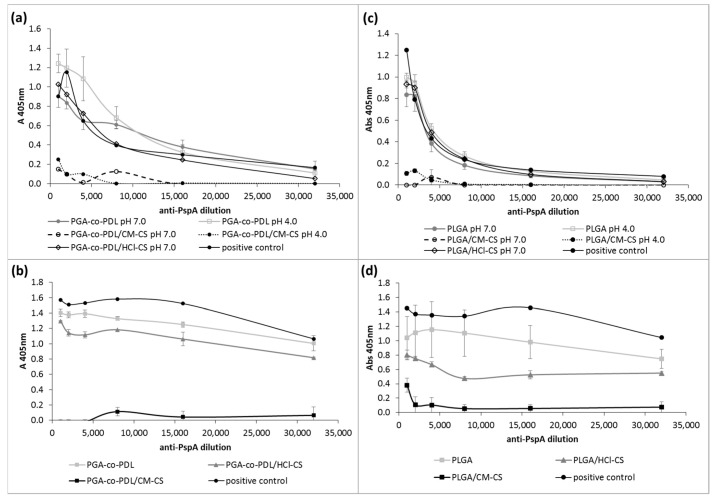
Recognition of PspA4Pro released from NPs by anti-PspA antibodies. (**a**) PGA-co-PDL and (**c**) PLGA NPs with or without chitosan prepared at pH 4.0 or 7.0 by *o*/*w* single emulsion with surface-adsorbed PspA4Pro. (**b**) PGA-co-PDL and (**d**) PLGA NPs prepared at pH 7.0 by *w*/*o*/*w* double emulsion with encapsulated PspA4Pro. Chitosan hydrochloride (HCl-CS) or carboxymethyl chitosan (CM-CS) were employed. PspA4Pro was used as a positive control. Data represent mean ± SD, n = 3.

**Figure 6 pharmaceutics-14-01238-f006:**
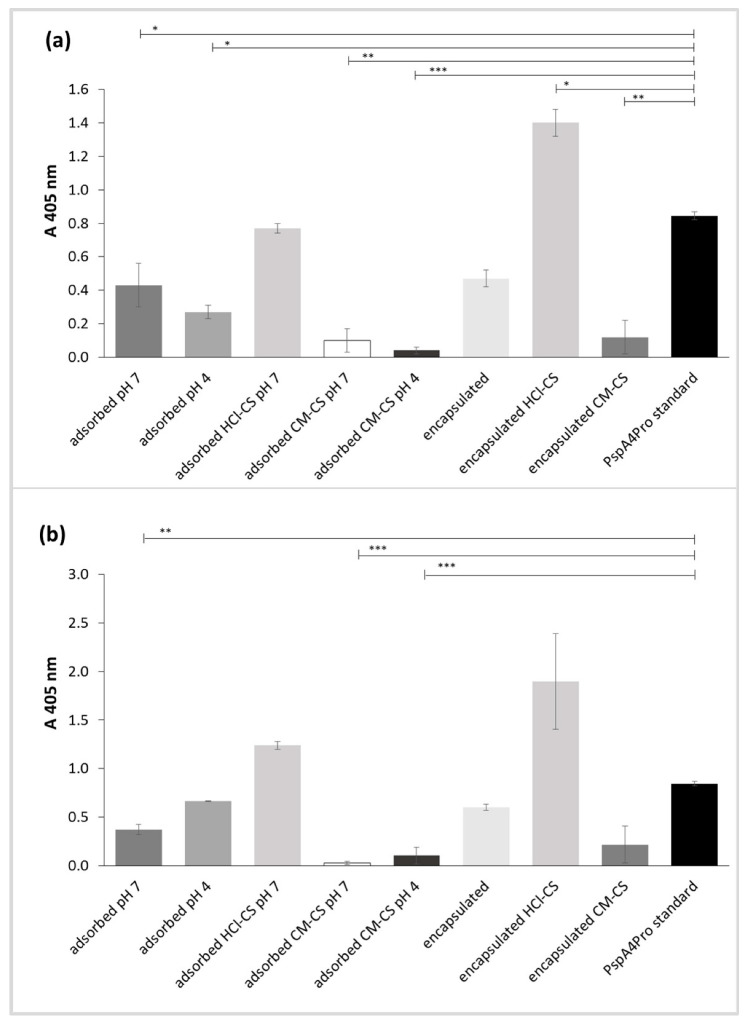
Lactoferrin binding assay of PspA4Pro released from NPs prepared with or without chitosan carrying surface-adsorbed PspA4Pro at pH 4.0 or 7.0 and NPs prepared at pH 7.0 by water-oil-water double emulsion with encapsulated PspA4Pro. (**a**) PGA-co-PDL NPs. (**b**) PLGA NPs. Chitosan hydrochloride (HCl-CS) or carboxymethyl chitosan (CM-CS) were employed. PspA4Pro standard was used as positive control. Data represent mean ± SD, n = 3. Statistical analysis: ANOVA/Tukey’s comparison test, * *p* < 0.05, ** *p* < 0.01, *** *p* < 0.001.

**Figure 7 pharmaceutics-14-01238-f007:**
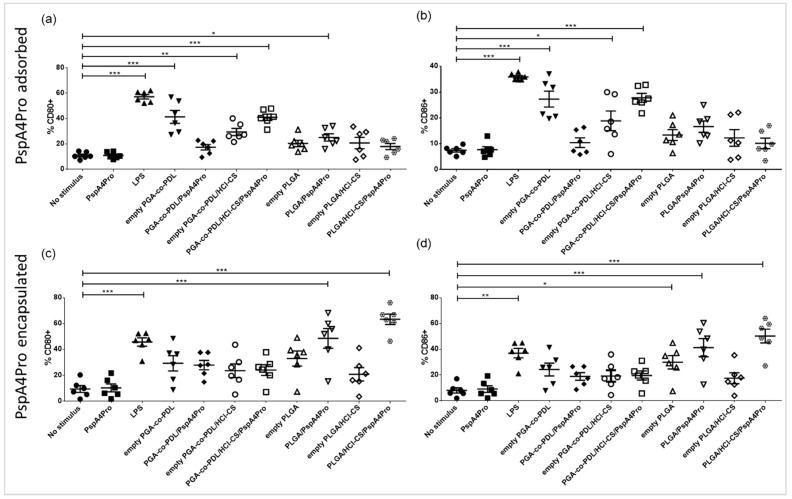
Activation of dendritic cells by NCMPs carrying different NP formulations containing adsorbed or encapsulated PspA4Pro. NPs were prepared with or without chitosan hydrochloride (HCl-CS) and contain PspA4Pro adsorbed onto the surface (**a**,**b**) or encapsulated into the NPs (**c**,**d**). Percentage of CD80+ DCs (**a**,**c**); percentage of CD86+ DCs (**b**,**d**). Significant differences in relation to the control group without stimulus are indicated (one-way ANOVA, Tukey’s comparison test). * *p* < 0.05, ** *p* < 0.01, *** *p* < 0.001.

**Figure 8 pharmaceutics-14-01238-f008:**
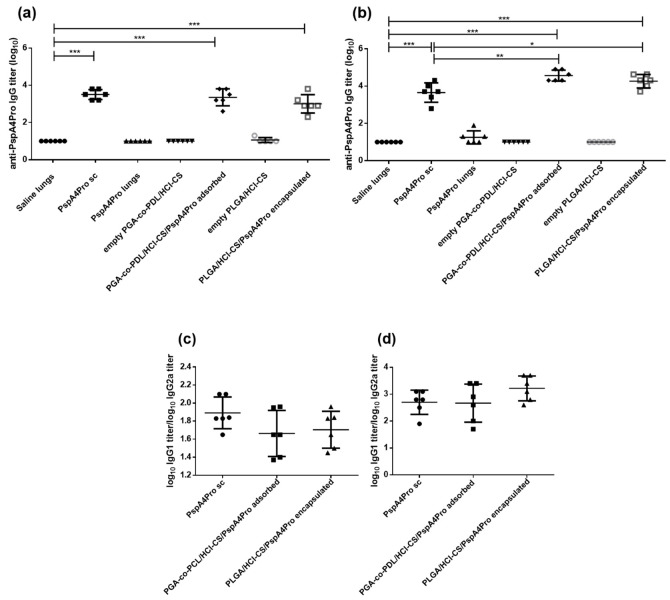
Induction of serum anti-PspA4Pro IgG antibodies by mucosal immunization targeting the lungs. Anti-PspA4Pro IgG antibodies in sera from mice inoculated with 2 doses of the indicated formulations containing 2 µg PspA4Pro (**a**,**c**) or 6 µg PspA4Pro (**b**,**d**) was determined by ELISA. Positive control mice were inoculated sc with PspA4Pro. Log_10_ anti-PspA4Pro IgG titers (**a**,**b**) and log_10_ anti-PspA4Pro IgG1 titer/log_10_ anti-PspA4Pro IgG2a titer ratios (**c**,**d**) are shown. Symbols represent each individual. Means ± standard errors are shown. Significant statistically differences are indicated (One-way ANOVA, Tukey’s multicomparison test for **a**,**b**; unpaired *t*-test for **c**,**d**). * *p* < 0.05, ** *p* < 0.01, *** *p* < 0.001.

**Table 1 pharmaceutics-14-01238-t001:** Conditions evaluated for optimization of nanoparticle preparation.

Condition	DCM ^1^ (mL)	PVA Volume in 1st Aqueous Phase (mL)	PVA Concentration in 1st Aqueous Phase (%)	Sonicator Amplitude Setting (%)	Time of Sonication (s)	PVA Concentration in 2nd Aqueous Phase (%)
1	1.0	4.0	2.5	65	60	1.0
2	1.0	5.0	10.0	45	300	1.0
3	1.5	3.0	2.5	45	120	1.0
4	1.5	3.0	2.5	60	60	1.0
5	1.5	4.0	5.0	65	300	0.5
6	1.5	4.0	5.0	65	300	0.5
7	2.0	4.0	10.0	45	120	1.0
8	2.0	3.0	10.0	65	120	0.5
9	2.0	5.0	10.0	65	300	0.75
10	2.0	5.0	10.0	65	120	0.75

^1^ DCM = Dichloromethane.

**Table 2 pharmaceutics-14-01238-t002:** Factors investigated for NP optimization.

Polymer	Chitosan	Antigen Location	pH
PGA-co-PDL	no chitosan	no antigen	4.0
PLGA	chitosan hydrochloride (HCl-CS)	adsorbed	7.0
	carboxymethyl chitosan (CM-CS)	encapsulated	

**Table 3 pharmaceutics-14-01238-t003:** Effect of chitosan hydrochloride or carboxymethyl chitosan addition on particle size, zeta-potential, and poly dispersity index (PDI) of hybrid NPs formulated by the *w*/*o*/*w* double emulsion method (mean ± SD, n = 3).

	Chitosan Hydrochloride (1 mg/mL)			Carboxymethyl Chitosan *		
	Size (nm)	Charge (mV)	PDI	Size (nm)	Charge (mV)	PDI
PGA-co-PDL	291.0 ± 9.2	17.4 ± 1.0	0.09 ± 0.04	280.5 ± 8.8	−20.1 ± 1.5	0.13 ± 0.02
PLGA	310.2 ± 6.0	13.2 ± 0.8	0.10 ± 0.03	299.4 ± 7.1	−40.5 ± 0.9	0.09 ± 0.07

* Carboxymethyl chitosan 1 mg/mL for PGA-co-PDL formulation and 0.5 mg/mL for PLGA formulation.

**Table 4 pharmaceutics-14-01238-t004:** Effect of pH and chitosan type on PspA4Pro adsorption and encapsulation into PGA-co-PDL and PLGA hybrid NPs prepared by single (*o*/*w*) or double (*w*/*o*/*w*) emulsion solvent evaporation method (mean ± SD, n = 3).

		PGA-co-PDL		PLGA	
	Formulation	PspA (µg)/NP (mg)	%LE *	PspA (µg)/NP (mg)	%LE *
Adsorption	without chitosan pH 7.0	25.57 ± 8.13	63.94 ± 20.33	8.24 ± 6.28	20.60 ± 15.7
	without chitosan pH 4.0	9.31 ± 4.34	23.26 ± 10.85	36.28 ± 0.23	90.69 ± 0.57
	HCl-CS pH 7.0	36.49 ± 1.15	91.23 ± 2.87	37.59 ± 1.94	93.97 ± 4.84
	CM-CS pH 7.0	15.41 ± 2.48	38.52 ± 6.21	10.55 ± 5.10	26.40 ± 12.62
	CM-CS pH 4.0	11.39 ± 4.13	28.49 ± 10.33	29.62 ± 7.80	74.04 ± 19.48
Encapsulation	without chitosan pH 7.0	4.14 ± 0.22	2.07 ± 0.80	29.57 ± 4.9	14.78 ± 7.36
	HCl-CS pH 7.0	22.88 ± 2.65	11.44 ± 6.62	59.33 ± 1.32	29.66 ± 4.65
	CM-CS pH 7.0	7.86 ± 1.83	3.93 ± 6.71	11.88 ± 4.15	5.94 ± 1.7

* %LE = load efficiency.

**Table 5 pharmaceutics-14-01238-t005:** Survival after lethal challenge with ATCC6303.

PspA4Pro Dose	Group	Alive/Total	% Survival	*p* *
2 µg	saline sc	0/6	0	-
	PspA4Pro lungs	0/6	0	-
	PspA4Pro sc	2/6	33.3	0.23
	Empty PGA-co-PDL/HCl-CS	0/6	0	-
	PGA-co-PDL/HCl-CS/PspA4Pro adsorbed	4/6	66.7	0.03
	Empty PLGA/HCl-CS	0/6	0	-
	PLGA/HCl-CS/PspA4Pro encapsulated	2/6	33.3	0.23
6 µg	saline sc	0/6	0	-
	PspA4Pro lungs	0/6	0	-
	PspA4Pro sc	2/6	33.3	0.23
	Empty PGA-co-PDL/HCl-CS	0/6	0	-
	PGA-co-PDL/HCl-CS/PspA4Pro adsorbed	5/6	83.3	0.007
	Empty PLGA/HCl-CS	0/6	0	-
	PLGA/HCl-CS/PspA4Pro encapsulated	6/6	100.0	0.001

* Fisher Exact Test.

## Data Availability

All data presented in this study are available in the article.

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
