# Peer review of "Pneumococcal Surface Protein A-Hybrid Nanoparticles Protect Mice from Lethal Challenge after Mucosal Immunization Targeting the Lungs"

_pharmaceutics, 2022, doi:10.3390/pharmaceutics14061238_

Round 1

Reviewer 1 Report

In this manuscript, the authors reported hybrid nanoparticles, PLGA and PGA-co-PDL, incorporated with chitosan absorbed PspA, for intranasal immunisation against lethal pneumococcal challenge. They also compared their IgG production and protection efficacy after immunisation. Their results indicate that 100% mice immunized with PLGA/HCl-CS/encapsulated-PspA were protected from lethal challenge, while PGA-co-PDL/HCl-CS/adsorbed-PspA protected 83% mice. Overall, the idea of nasal vaccination using the HCl-chitosan and PLGA based nanoparticles for pneumococcal challenge shows a certain degree of novelty, if the authors could improve the data organization. At current stage, the authors have included many parameters in the formulation preparation, ie, chitosan type, the way of formulate antigen with polymer nanoparticles, which could influence the immunity provoking efficiency and the final protecting efficacy. However, the data are not well organized and the included groups in the comparison are not consistent in several tests. They also failed to give explanation of the observed results. There are still some points need to be improved, as detailed below. I suggest not to accept this manuscript before major revision.

  1. As the authors compared the function of so many formulations, a table or a scheme would better demonstrate the different parameters of compared formulations and resultant functions.
  2. The authors should explain the role of chitosan in the hybrid formulation. In the introduction part, the authors mentioned “mucoadhesive substances as chitosan to improve residence time and antigen presentation at mucosal surfaces and facilitate the crossing of tight gap junctions between epithelial cells [8,9]” As shown in the results, chitosan-PspA encapsulated in the PLGA nanoparticles offer the highest protection compared to other groups. Is it still possible for chitosan to help nanoparticles adhere to mucus when chitosan is embedded in the PLGA nanoparticles, not decorated on the surface of nanoparticles?
  3. Except using SDS-PAGE to determine the integrity of antigen, the authors should use CD to investigate the secondary structure of PspA released from the NPs.
  4. In figure 2a, the results shows that when the HCl-CS concentration increased from 0.2 to 1, the particle size first increased a lot then decreased. Do the authors have any explanations to this phenomenon?
  5. In table 3, the authors compared the LE% and loading capacity of different formulations. But the table caption is a duplicate of table 2. Also, in adsorption section, different pH value were compared (pH7 and 4), why not keep it consistent in the encapsulation section?
  6. In figure 3, we can see the release of antigen reached platform after about 4 hours in all groups and hardly increase in the following testing hours. Dose it indicate there is something wrong in the determination of antigen loading amount? As the loading amounts were calculated indirectly (feeding amount – antigen in the supernatant), is it possible that some antigens have been degraded during the nanoparticle fabrication process? The authors should test the loading efficiency directly. For example, extract the loaded antigen from NPs. Also, the resolution in Figure 3 should be increased.
  7. The author should give some explanation to the difference in the DC activation stimulated by those hybrid nanoparticles.
  8. What is the purpose of comparing the lactoferrin of PsPA?

Author Response

We thank the reviewer for the very useful comments, which have allowed us to improve the manuscript. The detailed answers are in the attached file.

English language was reviewed by Prof. Saleem, a native English speaker.

Page and line numbers are given according to the revised ".pdf" version.

Reviewer 2 Report

This manuscript describes the production and characterisation of polymeric nanoparticles for incorporation and delivery of protein antigens from Pneumococcal surface protein A. It builds on previous work by the authors in this area.

A number of formulations are for the production of encapsulated or absorbed NPs are tested. A through physical characterisation of the NPs is carried out. Including the degree of protein loading and rate of release.

Further studies investigate structural changes to the protein which might alter the immunogenicity.

The best performing NPs are assayed in vivo for ability to protect mice against Pneumococcal infection.

The studies are well preformed, analysed and described.

The main finding being the identification of an NP formulation able to provide 100% protection when delivered nasally to the mice. This therefore has potential for further study as a vaccine candidate.

I think this is a worthy, interesting, and well conducted study. Worthy of publication.

In my opinion it requires only very minor modifications.

  1. In Section 3.5, figure 4 apparent structural changes to the antigen are reported, which reduce its ability to be recognised by existing antibodies. An additional band of ~100kDa is seen on gels. Is it likely that this is a dimer of the protein? Perhaps the authors could discuss further.
  2. Could the figures be reproduced at a higher resolution than the ones provided in the pdf please. In many of the graphics labelling appears blurred and hard to read.

Author Response

 We thank the reviewer for the compliments.

  1. In Section 3.5, figure 4 apparent structural changes to the antigen are reported, which reduce its ability to be recognised by existing antibodies. An additional band of ~100kDa is seen on gels. Is it likely that this is a dimer of the protein? Perhaps the authors could discuss further.

Answer: We agree that 100 kDa bands could be due to dimer formation, but it is not clear how carboxymethyl chitosan could contribute to PspA dimerization or aggregation. We add this information on section 3.5 of the revised version. (Section 3.5, page 11, lines 427-428, page and line numbers according to pdf file of the revised manuscript).

Reviewer 3 Report

The article entitled Pneumococcal surface protein A-hybrid nanoparticles protect mice from lethal challenge after mucosal immunization targeting the lungs is a document of interesting subject matter.
1. It is expected to have an extensive literature review followed by an in-depth and critical analysis of the state of the art, and identify challenges for future research in the Introduction.

  1. The authors should do the analysis the conclusion section must clearly establish a strong correlation with the proposed topic.
    3. Your abstract should clearly state the essence of the problem you are addressing, what you did and what you found and recommend. That will help a prospective reader of the abstract to decide if they wish to read the entire article
  2. The objective or objectives should be clearly elucidated in the last paragraph of the introduction.
  3. Try to create a purposeful relationship between paragraphs in the Introduction.

Author Response

We thank the reviewer for the helpful advices. The detailed answers are attached. 

Page and line numbers are given according to the revised ".pdf" version.

Round 2

Reviewer 1 Report

NA

Reviewer 3 Report

Authors addressed all comments completly.